# Independent Research on Cancer Pain Management in the Setting of Early Palliative Care: A Flywheel to Counteract General Opioid Misuse and Abuse

**DOI:** 10.3390/ijerph17197097

**Published:** 2020-09-28

**Authors:** Elena Bandieri, Leonardo Potenza, Fabio Efficace, Eduardo Bruera, Mario Luppi

**Affiliations:** 1Oncology and Palliative Care Units, Civil Hospital Carpi, USL, 41012 Carpi, Italy; e.bandieri@ausl.mo.it; 2Hematology Unit and Chair, Azienda Ospedaliera Universitaria di Modena and Department of Medical and Surgical Sciences, University of Modena and Reggio Emilia, 41125 Modena, Italy; leoanrdo.potenza@unimore.it; 3Italian Group for Adult Hematologic Diseases (GIMEMA), Data Center and Health Outcomes Research Unit, 00182 Rome, Italy; f.efficace@gimema.it; 4Palliative Care & Rehabilitation Medicine, UT MD Anderson Cancer Center, Houston, TA 77030, USA; ebruera@mdanderson.org

**Keywords:** cancer pain, early palliative care, guidelines, opioid

## Abstract

The increased recognition of the high prevalence and important burden of cancer pain and the documentation of a large proportion of patients receiving inadequate analgesic treatment should have reinforced the need for evidence-based recommendations. The World health Organization (WHO) guidelines on cancer pain management—or palliative care—are traditionally based on a sequential, three-step, analgesic ladder according to pain intensity: nonopioids (paracetamol or nonsteroidal anti-inflammatory drugs) to mild pain in step I; weak opioids (eg, codeine or tramadol) to mild-moderate pain in step II; and strong opioids to moderate-severe pain in step. III. Despite the widespread use of this ladder, unrelieved pain continues to be a substantial concern in one third of patients with either solid or hematologic malignancies. The sequential WHO analgesic ladder, and in particular, the usefulness of step II opioids have been questioned but there are no universally used guidelines for the treatment of pain in patients with advanced cancer and not all guideline recommendations are evidence-based. The American Society of Clinical Oncology and the European Society of Medical Oncology have recommended the implementation of early palliative care (EPC), which is a novel model of care, consisting of delivering dedicated palliative service concurrent with active treatment as early as possible in the cancer disease trajectory. Improvement in cancer pain management is one of the several important positive effects following EPC interventions. Independent well-designed research studies on pharmacological interventions on cancer pain, especially in the EPC setting are warranted and may contribute to spur research initiatives to investigate the poorly addressed issues of pain management in non cancer patients.

## 1. Introduction

Approximately one third of cancer patients still do not receive pain medication proportional to their pain intensity [1]. A recent rapid increase in opioid abuse and misuse has raised major concerns about lives’ cost, increased economic burdens, and related social issues [2]. One unintended consequence of the opioid fight against the epidemic has been the major decrease in daily opioid dose prescribed to patients with cancer pain by oncologists [3]. Based on a search of Cochrane Database of Systematic Reviews (January 2014 to January 2019), complemented by a MEDLINE search in mid 2019, a recent review of eight national and international guidelines on pain management in adult cancer patients, has highlighted that a large majority of interventions commonly recommended by guidelines are not always supported by a robust evidence base [4]. In our opinion, this is particularly true for the management of moderate cancer pain. A Cochrane systematic review shows that there is only little, low-quality evidence for the use of codeine in cancer pain [5]. In an early retrospective study by Ventafridda and colleagues [6], the effectiveness of step II of the World Health Organization (WHO) method had a time limit of 30 to 40 days and, for most patients, the shift to step III was made mainly because of inadequate analgesia rather than adverse events. In daily clinical practice, step II had often been bypassed in favor of strong opioids, although the strategy had not been supported by strong scientific evidence, because it had been investigated by only two randomized controlled studies (RCTs) enrolling only 92 and 54 terminally ill patients, respectively [7,8] and one prospective study [9]. Moreover, these three studies had reported inconclusive results because of the low number and representativeness of the patient sample and the low statistical power, leading to a weak recommendation for either a step II opioid or low doses of a step III opioid, as an alternative, in international guidelines [10,11,12]. Afterwards, a 28-day, open-label RCT was conducted by our fully independent research group with no external financial support and included 240 adults with moderate cancer pain who were assigned to receive either a weak opioid (i.e., tramadol alone or in combinations with paracetamol, or codeine in fixed combination with paracetamol,) or low-dose morphine [13]. In this study, the number of responder patients, defined as patients with a 20% reduction in pain intensity on the numerical rating scale occurred in 88.2% and in 57.7% of the low-dose morphine and weak-opioid group, respectively (odds risk, 6.18; 95% Confidence Interval (CI), 3.12 to 12.24; *p* < 0.001). The percentage of responder patients was higher in the low-dose morphine group, as early as at 1 week, and clinically meaningful (≥30%) and highly meaningful (≥50%) pain reduction from baseline was significantly higher in the low-dose morphine group (*p* < 0.001). A change in the assigned treatment occurred more frequently in the weak-opioid group, because of inadequate analgesia. Also, the general condition of patients, based on the Edmonton Symptom Assessment System overall symptom score, was better in the morphine group and adverse effects were similar in both groups [13]. Our study provided the first evidence-based information that, although step II opioids are indeed effective when used for limited time intervals, low-dose morphine can be usefully anticipated and can substitute for weak opioids in patients with cancer and moderate pain, because of greater efficacy and a comparable toxicity profile [13]. Unfortunately, no further studies neither confirmed nor disproved the results of our RCT. One study, in which the effects of a two-step approach with opioids are compared with the effects of a standard three-step approach (ClinicaTrialas.gov NCT01493635) started in November 2012 and was completed about four years later, but no information is yet available in the scientific literature [14]. Despite the limited available evidence mentioned above, randomized controlled studies were performed to compare the analgesic effect of strong opioids on patients with cancer pain, including not only patients with severe but also with moderate pain, as reviewed in a recent meta-analysis [15]. Consistent with this, the last WHO guidelines again discussed the meaning of the three-step analgesic ladder [16], which had been introduced in 1986 and disseminated worldwide to help medical doctors and health-care professionals to use a few analgesics appropriately. The assessment that “A cancer pain management ladder is useful as a teaching tool and as a general guide to pain management based on pain severity but it cannot replace individualized therapeutic planning based on careful assessment of each individual patient’s pain…” [16] indirectly reflects, in our opinion: (a) the general clinical notion and practice that the second step of the three-step analgesic ladder may be skipped; and (b) the absence of further evidence-based results from more than one RCT addressing this otherwise relevant issue. Of interest, for mild to moderate cancer pain, the last European Society of Medical Oncology (ESMO) guideline recommended the use of the drugs of the second-step WHO analgesic ladder with a level of evidence and grade of recommendation of III, C, respectively, but also acknowledged the use of low doses of strong opioids as an alternative to weak opioids, based on the results of the only available RCT [13], with a level of evidence and grade of recommendation of II, C, respectively [17]. Similarly, the last National Comprehensive Cancer Network (NCCN) guideline recognized the use of strong opioids for moderate cancer pain, with a NCCN category and consensus of 2 A (i.e., based upon lower-level evidence, but for which there is uniform NCCN consensus that the intervention is appropriate) [18].

In conclusion, based on a revision of several guidelines, and especially the most recent ESMO, NCCN, and WHO guidelines [16,17,18], showing that most of the recommendations on cancer pain management are based upon a low-level of evidence, we believe that there is urgent need for independent research studies, which should be designed and funded, hopefully, also under the support and/or endorsement of health care systems, to upgrade the level of evidence and the grades of recommendations.

One of the possible reasons for the successful completion of our RCT study, in due time [13], may reside in the clinical context of the study itself, namely the early palliative care (EPC) setting. Indeed, more than half of enrolled cancer patients in our study [13] were receiving active antitumor therapy, while patients enrolled in the two previous studies [7,8] consisted of patients at the end of life phase care, and this might have probably contributed to the low recruitment rate. Cancer patients with advanced diseases, assigned to receive EPC integrated with standard care, compared with patients assigned to standard oncologic care alone, reported improvements in quality of life, lower depression, lower rates of chemotherapy use near death, as well as longer enrolments in hospice care, superior awareness of prognosis, and longer survival [19]. Both the American Society of Clinical Oncology [20,21] and the ESMO [22] have recognized that patients with advanced cancer should receive dedicated palliative care services, concurrent with active treatment, early in their disease trajectory, namely within eight weeks since diagnosis of advanced/metastatic cancer. Our research group showed that EPC integrated with primary oncologic care was an independent factor associated with a 31% reduced risk of suffering from severe pain [23]. It is conceivable that the EPC setting may represent an ideal setting to design and complete independent research studies on cancer pain management. Actually, difficulties of conducting multicenter trials of EPC interventions in advanced cancer patients have recently been described [24] and research studies should be designed to compare different models of EPC intervention in different clinical and regional/geographical situations. Moreover, more efforts should also be done to implement education programs for medical and nurse students, as early as possible, which could be continued and refined in the following specialty school courses, including specific medical communication teaching courses [25,26].

## 2. Conclusions

We think that the implementation and dissemination of evidence-based knowledge on the use of drugs in cancer pain management, especially in the favorable and clinically relevant context of EPC interventions, may be a pragmatic and effective way to counteract the opioid misuse and abuse, in the oncology and hematology wards and among patients and their caregivers. The further dissemination of scientific results, leading to a robust knowledge on cancer pain management, may, in turn, spur a broad scientific interest and multiple novel research initiatives and well-designed studies on the management of pain in non-cancer patients and contribute to generate new insights on such themes either in the scientific community or in the clinical care wards or in the general population.

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
