# Peer review of "Independent Research on Cancer Pain Management in the Setting of Early Palliative Care: A Flywheel to Counteract General Opioid Misuse and Abuse"

_ijerph, 2020, doi:10.3390/ijerph17197097_

Round 1

Reviewer 1 Report

I find the article very interesting and I congratulate the authors. However, the title seems inappropriate . The article is more about the benefits of early palliative care than about preventing opioid misuse. I suggest that the title be changed, although the last paragraph is kept referencing its potential benefit of palliative care in the adequate use of opioids

Author Response

I would like to thank the reviewer for the positive evaluation. I would dare not to change the title which indeed includes the reference to opioid misuse and abuse. I wonder whether the Editor may provide suggestions or definitely agrees with our proposal

Reviewer 2 Report

I suggest the authors to contact an expert of english medical writing, as I noticed several mistakes of spelling ( ex: opiopid in Title section) and of phrases across the text.

Methods are well described, but I mean that the review could be more complete if the authors consider more databases (Cochrane and MEDLINE) before stating that few studies have been conducted on this topic (only 2 RCTs).

Author Response

We would like to thank the reviewer for the positive evalutation and for his/her suggestions.

We would like to thank the reviewer for the positive evaluation, the comments and suggestions.

The text has  extensively been revised to avoid mispellings.

We have included a  new reference (number 15) which is a meta-analysis of seven RCTs including patients with moderate to severe cancer pain. However, these studies compared the analgesic effects of different strong opioids and, indeed, evaluated  the impact of the three step WHO ladder not only in patients with severe but also in patients with moderate cancer pain. Again a comparison between weak and strong opioids was not performed.

The added text, briefly raising the above mentioned issue is in bold on page 3 line 6.  

Reviewer 3 Report

As this evolves would hope the authors would include a Pain Medicine provider who has a focus and interest in cancer pain

Author Response

We would like to thank the reviewer for the positive comment

Reviewer 4 Report

The authors pointed out that a large majority of interventions commonly recommended by clinical guidelines are not always supported by a robust evidence base, among them the management of moderate pain. In particular, the step II of the Who ladder is often bypassed in favor of strong opioids, cause patient experiences pain.

This topic is relevant and meaningful especially for those cancer patients not followed by palliative care services.

The argument itself is not original but an unequivocal and shared indication, as well described by the authors, has not yet been defined. I believe that this paper adds useful elements to the current scientific debate.

In my opinion the paper is clear and well written.

The authors argue in detail that most of the recommendations on cancer pain management are based upon low-level of evidence, the suggest that there is need for independent research studies, especially for patients assigned to simultaneous care or early palliative care.

The authors suggest that early palliative care is an optimal area of study but give little indication on how to study it. This is the weakest part of the paper. They pointed out that the concrete possibility to do research in this field is conditioned by the low number of patients included in EPCs and organizational models.

Author Response

We would like to thank the reviewer for the positive comment. Early Palliative Care (EPC) is, unfortunately, a neglected area of research and our simple message is not to limit studies addressing cancer pain management to end-of-life patients but to include  advanced cancer patients fololwed by an EPC team with the expertise in symptom management